# CAN WE TRUST THE ATTRIBUTION METHOD?

## ABSTRACT

Attribution methods are essential for interpreting deep learning models, helping to align model decisions with human understanding. However, their trustworthiness remains uncertain. Previous work has highlighted several design flaws in attribution methods such as the choice of reference points and the selection of attribution paths, but we argue that even a theoretically perfect attribution method—one that provides the true ground truth—cannot fully resolve the trust issue. For the first time, we summarize the specific manifestations of such issue: Two samples with infinitely close distances but different classification results share the same important feature attention region. We rigorously derive this phenomenon and construct scenarios demonstrating that attribution trust issues persist even under ideal conditions. Our findings provide a new benchmark for evaluating attribution methods and highlight the need for cautious application in real-world scenarios. Our code is available at: `https://anonymous.4open.science/r/Distrust-8677/`.

## 1 INTRODUCTION

In the field of interpretability in machine learning, attribution methods aim to measure the contribution of input features to model decisions, thereby enhancing model transparency and interpretability (Carvalho et al., 2019). With the widespread application of deep learning in high-risk domains, its "black-box" nature has generated a strong demand for decision interpretability. Attribution methods provide an intuitive understanding of model behavior and hold significant applications in critical fields such as medical diagnosis (Graziani et al., 2020), financial risk control (Tarashev et al., 2016), and autonomous driving (Shi et al., 2024). However, the robustness and stability of attribution methods remain challenging, particularly regarding their manipulability in high-dimensional data and complex neural network structures. Therefore, research on the reliability and generalization capability of attribution methods constitutes a crucial direction in the field of machine learning interpretability (Linardatos et al., 2020).

In practical applications, we employ attribution methods to examine the interpretability performance of deep learning models. Generally, trust between humans and models can be established by comparing the model's attention regions with human-understandable patterns. Dombrowski et al. (2019) have demonstrated that, given knowledge of the inner workings of interpretability algorithms, model explanations can be manipulated by imperceptible perturbations in input samples, suggesting inherent flaws in attribution algorithms. Although many attribution methods attempt to mitigate such deficiencies by adhering to more rigorous attribution axioms (Zhu et al., 2024a;c), this can still lead to trust crises. While continuous refinement of algorithmic design can improve the credibility of attribution, to avoid this cat-and-mouse game, our paper will discuss the issue of attribution trust beyond algorithmic design flaws. Admittedly, trust itself is an inherently abstract concept. Here, we provide a slightly more concrete analogy for the notion of "trust": just as we fear the possibility of vehicle malfunction and thus worry about safety when informed of potential brake loosening, similarly, we must highlight that interpretability results derived from attribution algorithms may yield counterintuitive conclusions under certain inevitable (and theoretically demonstrable) scenarios. Therefore, attribution outcomes should always be interpreted cautiously and critically, given these intrinsic limitations.

Suppose we have obtained a "perfect" attribution algorithm through continuous iterations, one that can provide ground truth attribution results. For two samples that are infinitely similar—typically perturbed by an imperceptible amount within the limits of computational precision—yet classified

into different labels, if their ground truth attribution results fall within the same region, then despite the explanation aligning with the ground truth, we still cannot fully trust this attribution algorithm, because it fails to capture the decisive factors that lead to differing classification outcomes.

Based on the above discussion, we summarize the specific manifestation of the attribution trust issue as follows:

***Two samples with infinitely close distances but different classification results share the same important feature attention region.***

Here, one might fall into a logical misconception—specifically, the empiricist fallacy—which assumes that two nearly identical samples naturally produce identical attribution results due to their close proximity. This intuitive reasoning overlooks a critical assumption of attribution methods: the two indistinguishable samples may belong to entirely different classes. For example, in Figure 1, two visually indistinct samples exist, one classified as a dog and the other as a cat. Attribution heatmaps, however, aim to highlight the essential features of each respective class. Remarkably, despite their differing labels, these nearly identical samples yield the same attribution results (later in our paper, we formally demonstrate that these attributions correspond precisely to the same ground truth).

Moreover, a prevalent viewpoint is that an ideal attribution algorithm should detect subtle differences, such as a minor pixel change altering a classification. While correct and desirable, this also introduces a conceptual paradox: the feature identified as crucial for transforming a "dog" into a "cat" is simultaneously essential for distinguishing a "cat" from a "dog," forming an equivalent dual problem. Thus, attribution methods correctly capture subtle differences, yet yield effectively identical explanations across different classes, directly undermining interpretability and trustworthiness. For instance, in tumor prediction, attribution highlighting the same region regardless of the presence or absence of a tumor poses a fundamental question: does interpretability genuinely enhance trust, or does it unintentionally confuse user decision-making?

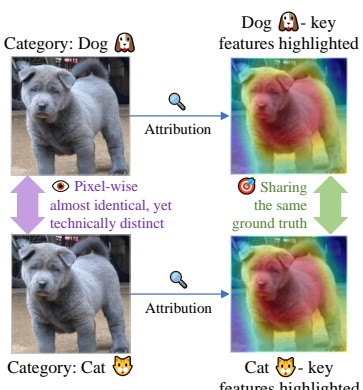

Figure 1: It's a dog! Because of the features in the top-right! No wait—it's a cat! Because of the features in the bottom-left! Strangely enough, both explanations assign exactly the same importance. Should we really trust attributions like this?

In our paper, we formally demonstrate that no matter how an attribution algorithm overcomes its design flaws to achieve a "perfect" ground truth explanation, such samples inevitably exist, implying that attribution distrust is an inherent issue. Furthermore, these samples must be meaningful in real-world scenarios rather than chaotic, random samples. Of course, beyond illustrating the trust issue, the ground truth explanation of attribution can also serve as an evaluation criterion to assess whether an algorithm produces high-quality attribution results. We argue that although attribution algorithms face trust issues, this does not render them entirely useless. Attribution methods still play a crucial role in the current field of deep learning and have many valuable applications (Dombrowski et al., 2019; Zhu et al., 2024b; Pan et al., 2021; Zhu et al., 2024d; Sundararajan et al., 2017; Wang et al., 2021; Kapishnikov et al., 2021). Therefore, refining attribution algorithms remains necessary; however, we must rigorously consider their scope of applicability. Our main contributions are as follows:

- We identify an inherent trust issue in the design of attribution algorithms and theoretically prove that this phenomenon is inevitable, highlighting a fundamental limitation that serves as a cautionary insight for the field of interpretability research.

- We rigorously derive and provide the ground truth attribution results to illustrate the trust issue, establishing them as a potential evaluation criterion for attribution algorithms.

- We conduct extensive experiments to substantiate our findings and make our implementation codes publicly available to facilitate further research.

## 2 RELATED WORK

Current interpretability methods fall into two main categories: early gradient-based saliency analysis and gradient-based attribution methods. The former uses model gradient information to assess input feature impact on outputs. Simonyan et al. (2013) pioneered this approach with Saliency Map (SM), which determines pixel influence on predictions by computing gradients of model output with respect to input. However, SM produces noisy results and suffers from gradient vanishing and oscillations. To address these issues, Springenberg et al. (2014) introduced Guided Backpropagation (GBP), which allows only positive gradients to pass through, reducing irrelevant feature influence and generating smoother results. Yet by altering standard gradient propagation, GBP may cause interpretations to deviate from actual neural network computations. Rather than direct gradient computation, Bach et al. (2015) proposed Layer-wise Relevance Propagation (LRP), which attributes relevance by recursively tracing neuron contributions within the network. Compared to SM, LRP better aligns with neural network computational structure and provides more stable results, though it requires selecting different propagation rules (such as the $\epsilon$-rule or $\alpha\beta$-rule) based on network architecture, increasing method complexity.

As neural networks deepen, gradient-based saliency methods struggle with smooth gradient variations, reducing interpretability. Sundararajan et al. (2017) proposed Integrated Gradients (IG), which mitigates gradient oscillations by computing gradient integrals along a path from baseline input (typically zero or average) to actual input. IG offers two advantages over methods like Saliency Maps and Layer-wise Relevance Propagation: the Sensitivity Axiom ensures features with significant prediction impact receive higher attribution values, and the Implementation Invariance Axiom maintains consistent results across functionally equivalent networks. However, IG suffers from high computational cost, noise sensitivity, and baseline selection difficulties. To improve efficiency, Hesse et al. (2021) proposed Fast Integrated Gradients (Fast-IG), which uses adaptive sampling to prioritize input regions with greater prediction impact, reducing computational overhead while maintaining accuracy. Addressing noisy gradients, Kapishnikov et al. (2021) developed Guided Integrated Gradients (GIG), which employs weighting functions to guide integral computation toward relevant regions, smoothing gradient variations and improving attribution robustness.

To select an appropriate attribution reference point, Wang et al. (2021) proposed the Boundary-based Integrated Gradients (BIG) method, which, for the first time, utilizes the nearest adversarial sample to the original input as the reference point and performs integration between this point and the target input to compute attributions. However, BIG defines the neural network's decision boundary using a linear attribution path, while neural network decision boundaries are typically nonlinear. As a result, BIG may fail to accurately capture the true attribution mechanism of the model. To address this limitation, Zhu et al. (2024b) introduced the AttEXplore method, which optimizes decision boundaries through model parameter exploration and transferable adversarial attacks. This approach enables attribution methods to effectively traverse multiple decision boundaries, reducing overfitting and improving generalization capability. The More Faithful and Accelerated Boundary-based Attribution (MFABA) method (Zhu et al., 2024c) incorporates second-order gradient information by approximating the Hessian matrix using a second-order Taylor expansion, thereby enhancing the precision of attribution computations. Additionally, MFABA employs an adaptive gradient ascent strategy, allowing adversarial samples to cross decision boundaries along the steepest gradient ascent direction rather than following a fixed linear path. Furthermore, the Iterative Search Attribution (ISA) method (Zhu et al., 2024a) introduces an iterative search mechanism that dynamically adjusts the attribution search path by combining gradient ascent and gradient descent. This iterative adjustment improves the accuracy of capturing key feature contributions in the model's decision-making process.

Despite significant progress in attribution methods for deep learning interpretability, their reliability remains problematic. Bilodeau et al. (2024) theoretically showed that widely used attribution methods may perform no better than random guessing when satisfying completeness and linearity properties. Completeness requires attribution sums to equal the difference between model outputs for given and baseline inputs, but this overlooks that neural networks may exhibit different attribution patterns across regions. Linearity mandates that attribution methods preserve additivity for linear model combinations, which aids mathematical tractability but neglects deep networks' highly nonlinear behavior. Consequently, attribution methods may assign importance to irrelevant features or misleadingly suggest causal significance when decision boundaries are strongly nonlinear.

Additionally, Dombrowski et al. (2019) argued that high-dimensional geometric properties make interpretability methods vulnerable to manipulation, as attribution methods rely on local linear approximations that become highly sensitive to small perturbations in high-dimensional spaces, highlighting their lack of adversarial robustness. While existing studies focus on attribution algorithm design flaws, our paper takes a different perspective: even if all design problems were resolved, attribution methods would still face an inherent trust issue.

# 3 METHOD

## 3.1 PROBLEM DEFINITION

Here, we define the notations for attribution. Let $f$ denote a neural network, and $f_A$ represent the confidence score of class $A$. The input is given by $x \in \mathbb{R}^n$, with label $y$, and the loss function is denoted as $L$. $C(x) = \arg \max_{1 \leq k \leq K} f_k(x)$ represents the class with the highest output confidence.

The purpose of attribution is to calculate the attribution value of each dimension on the sample. A higher attribution value for a particular dimension indicates its increased importance in determining the model's output.

## 3.2 DISTRUST ISSUES

The key to exposing the trust issue lies in constructing two samples that share an identical attention region yet yield entirely different classification results. These samples must be indistinguishable to human perception, which implies that their distance must be sufficiently small—potentially infinitesimally small. Under such conditions, even when using a correct attribution algorithm, it becomes impossible to distinguish between the attribution results of these two samples, which ultimately leads to the trust issue. Here, by identical attention region, we specifically mean that the two samples assign the same importance scores to the same set of features. To quantify this similarity, we compute attribution differences under three distance metrics—Manhattan distance, Euclidean distance, and Cosine distance—whose details can be found in Section 4.2. Building upon the previous discussion and the example illustrated in Fig. 1, we formally define the trust issue as follows:

***Given two samples with entirely different classification results yet infinitely close distances, any correct attribution algorithm will yield identical attribution results. This implies that the attribution outcome cannot serve as a reliable explanation for the model's decision and, therefore, cannot be trusted.***

In investigating the trust issue, it is crucial to ensure that all referenced samples hold practical significance rather than being arbitrary noise samples. If we adopt a rigorous mathematical definition, all examined samples must reside within $B_\epsilon(x)$, where $B_\varepsilon(x) = \{\tilde{x} \mid \|\tilde{x} - x\| \leq \varepsilon\}$. And $\epsilon$ is typically a small value, requiring only a few pixels for images. The term $P(x)$ denotes the true distribution of input samples. In practical discussions, any training or test sample can be selected, and the analysis can be conducted within its neighborhood, as these samples necessarily belong to the real sample distribution.

The most idealized assumption is that we can create the most accurate attribution algorithm, as our goal is to capture the true intent of the model—identifying the input features that are genuinely important for the model's decision-making process. Unfortunately, we cannot guarantee that an attribution algorithm is absolutely correct. We can only rely on fidelity metrics such as the insertion score and deletion score (Petsiuk et al., 2018) to determine whether one attribution algorithm is superior to another. However, by adopting this assumption, we aim to articulate the central argument of this paper: even if we obtain the most accurate attribution algorithm, the trust issue will still persist. This remains an unavoidable challenge in the study of deep learning interpretability. To rigorously substantiate the trust issue, we first present the core theoretical foundation of this paper.

***Theorem 1:*** Given a model $f$, a sample $x$, and a sample perturbation $\Delta x$, such that $C(x) \neq C(x + \Delta x)$. Then for $\forall \varphi > 0, \exists x_1, x_2 \in [\gamma(t) \mid t \in (0, 1)]$ satisfying $C(x_1) \neq C(x_2)$ and $\|x_1 - x_2\| < \varphi$.

where $\gamma(t) = x + \alpha(t)\Delta x, \gamma(0) = x$ and $\gamma(1) = x + \Delta x$. $\alpha(t)$ is a continuous monotonic function. The condition $x_1, x_2 \in [\gamma(t) \mid t \in (0, 1)]$ ensures that both $x_1$ and $x_2$ lie on the path connecting

$x$ and $x + \Delta x$, thereby guaranteeing that $x_1$ and $x_2$ belong to $B_\varepsilon(x)$. Additionally, the condition $\|x_1 - x_2\| \leq \Delta x$ implies that by continuously constructing $x_1$ and $x_2$, the distance between $x$ and $x + \Delta x$ can be progressively reduced.

***Proof of Theorem 1:*** In practice, a typical neural network can be viewed as a finite composition of "linear mappings plus activation functions." Because the finite composition of continuous functions remains continuous, such a neural network is essentially built from continuous functions through a limited set of fundamental continuous operations—namely addition, multiplication, and composition—thus maintaining continuity. In mathematical terms, to rigorously establish the continuity of such a neural network, one typically employs the $\varepsilon$-$\delta$ definition from real analysis or topology. For a function $f : \mathbb{R}^n \to \mathbb{R}^K$, continuity over $\mathbb{R}^n$ implies that for any $x \in \mathbb{R}^n$ and for any $\varepsilon > 0$, there exists some $\delta > 0$ such that $\|x' - x\| < \delta \implies \|f(x') - f(x)\| < \varepsilon$. Due to the continuity of the neural network, there exists a $t^*$ such that $\gamma(t^*)$ lies on the decision boundary. In other words, at $\gamma(t^*)$, at least two output components coincide, leading to a switch in the maximally activated component. In the neighborhood of $t^*$, let us define $t_1 = t^* - \eta$ and $t_2 = t^* + \eta$. As long as $\eta$ is sufficiently small and satisfies $0 < t^* - \eta < t^* + \eta < 1$, we then obtain $x_1 := \gamma(t_1)$, $x_2 := \gamma(t_2)$. Because the trajectory crosses the decision boundary, it necessarily follows that $C(x_1) \neq C(x_2)$. Meanwhile, as $\varepsilon \to 0$, $\|x_1 - x_2\|$ approaches 0, implying that it can be made arbitrarily small. Therefore, we can readily derive that $\|x_1 - x_2\| = \|\gamma(t_1) - \gamma(t_2)\| = |\alpha(t_1) - \alpha(t_2)| \cdot \|\Delta x\| \leq \|\Delta x\|$.

Consequently, from a theoretical standpoint, we can indeed construct **two semantically meaningful samples whose pairwise distance becomes arbitrarily small yet still yield differing classification outcomes** (completely indistinguishable to the human eye). The existence of $x$ and $x + \Delta x$ serves to confine the construction of $x_1$ and $x_2$ within $B_\varepsilon(x)$, thereby ensuring that both $x_1$ and $x_2$ remain semantically meaningful. To generate $x$ and $x + \Delta x$, one can leverage adversarial attack strategies (Goodfellow et al., 2014):

$$x_t = x_{t-1} + \eta \cdot \text{sign} \frac{\partial L(x_t, y)}{\partial x_t} \tag{1}$$

Eq. 1 specifies the update rule for $x$, where $\eta$ denotes the attack learning rate and $x_0 = x$. Suppose the adversarial attack completes in $T$ steps, meaning $C(x_T) \neq C(x_0)$. We then define $\Delta x = x_T - x_0$. To ensure $x_T$ remains within $B_\varepsilon(x)$, we must choose $\varepsilon \geq \eta \cdot T$. Note that this construction is not unique—any method that satisfies these conditions for $\Delta x$ would also suffice.

After obtaining $x$ and $x + \Delta x$, we can construct $x_1$ and $x_2$ from $x_{T-1}$ and $x_T$, respectively. In this setup, $\|x_1 - x_2\|_\infty = \eta$. By making the attack learning rate $\eta$ arbitrarily small, the difference between $x_1$ and $x_2$ inevitably approaches zero. This reduction process is always achievable: for example, one can scale $\eta$ by $\frac{\eta}{S}$ and perform $S$ attack steps to reach the same goal, where the update rule becomes $x_t = x_{t-1} + \frac{\eta}{S} \cdot (x_T - x_{T-1})$. During these steps, at least one will succeed in reaching the adversarial objective. Iterating this procedure repeatedly shrinks $\eta$, ultimately yielding an arbitrarily small gap between $x_1$ and $x_2$. In practice, we often adopt a smaller $\eta$ from the outset to bypass the repeated scaling stage.

Next, we show that these two samples share the same ground-truth attribution result and provide the method used to compute this ground truth. We begin with an intuitive explanation of why their attributions coincide. Consider two infinitesimally close samples whose classifications differ; their only difference arises from the residual $x_2 - x_1$ (or equivalently $x_1 - x_2$), which yields the same outcome in both cases. These features contained in the residual are critical for both classes: removing any of these features from either class induces a shift in its category. In other words, the fewer the features that need to be altered to change the classification, the more essential those features must be. When this change is infinitesimally small, it highlights that these features are of the highest importance. To illustrate this more intuitively, consider a scenario where a sample transitions from class A to class B due to changes in 10 feature dimensions. For clarity, we simplify the discussion by focusing solely on the feature dimensions, temporarily ignoring the magnitude of changes. Suppose further analysis reveals that modifying only 3 of these dimensions is sufficient to induce the class change. This implies that the remaining 7 dimensions are not essential, as they are not necessary for the transition. By extension, features whose influence diminishes in the infinitesimal limit can be considered non-informative. In contrast, the minimal set of features that retains discriminative power under infinitesimal perturbations constitutes the most critical components for the classification decision.

Let us denote $r = x_2 - x_1$ as the residual shifting $x_1$ to $x_2$. Let $A$ be the class of $x_1$ and $B$ the class of $x_2$. From this definition, $\|r\|_\infty$ approaches an infinitesimal value; hence, it is precisely this

negligibly small residual $r$ that transitions the model's output from class $A$ to class $B$. *Because the perturbation is vanishingly small, there always exists a neighborhood where a first-order Taylor approximation is reliable and higher-order effects are negligible; this local linearity is the basic assumption behind gradient descent in neural networks and is also the locality premise used by surrogate methods such as LIME.* Concretely, for any sufficiently smooth scalar functional $R$ we use

$$R(x + \Delta x) = R(x) + \Delta x \cdot \frac{\partial R(x)}{\partial x} + \mathcal{O}(|\Delta x|^2), \tag{2}$$

*where $\mathcal{O}(|\Delta x|^2)$ denotes higher-order infinitesimals that vanish faster than $|\Delta x|$ as $|\Delta x| \to 0$.* Moreover, due to the infinite proximity mentioned in the theory above, $r$ serves only this single task, thereby ignoring any changes in the outputs of other classes. Unless all other classes are entirely correlated with the class needs to be changed—a condition that is practically impossible—such cases are bound to arise.

*We apply the first-order expansion in Eq. equation 2 with $\Delta x = r$ at the relevant base point.*

Since the role of $r$ is to lower the confidence in class $A$ while boosting the confidence in class $B$, we define $R(x) = f_A(x) - f_B(x)$. (To avoid ambiguity, while operations like softmax consider other class confidences, they are treated as internal mechanisms, and our analysis remains independent of such details.) Our aim is to construct a ground-truth attribution that captures the influence of different components in $r$ on $R(x)$. To that end, we derive the following equation:

$$R(x_2) - R(x_1) = r \cdot \frac{\partial R(x_1)}{\partial x_1} = -r \cdot \frac{\partial R(x_2)}{\partial x_2} = \sum_{i=1}^{n} r_i \frac{\partial R(x_1)}{\partial x_1(i)} = \sum_{i=1}^{n} \left( -r_i \frac{\partial R(x_2)}{\partial x_2(i)} \right) \tag{3}$$

In particular, $r_i \cdot \frac{\partial R(x_1)}{\partial x_1(i)}$ can be viewed as the attribution score for the $i$-th dimension, and the ground-truth computation process adheres to the attribution axioms introduced in (Sundararajan et al., 2017). In Eq. 3, the sum of per-dimension attributions within $r$ equals the difference in $R(x)$, thereby indicating how $r$ contributes to the shift in the class output. *This approximation becomes increasingly accurate as the distance between $x_1$ and $x_2$ tends to zero because the neglected term in Eq. equation 2 is $\mathcal{O}(|r|^2)$, i.e., a higher-order infinitesimal.* At this scale, higher-order infinitesimals can be omitted, **which is the basic assumption underlying gradient descent and is widely adopted in neural networks.** Because our perturbation distance is vanishingly small, one can always identify a region where this approximation holds reliably.

In this context, the only assumption underlying this process is that the model satisfies a local first-order approximation. This assumption is even weaker than that required by gradient descent during model training, where the model is expected to exhibit first-order behavior over the scale of the learning rate. In contrast, our assumption holds only in an infinitesimally small neighborhood. Moreover, to avoid potential misunderstandings, it is important to emphasize that this gradient-based assumption is significantly weaker than those typically made in gradient-based attribution methods. In such methods, gradients are assumed to be meaningful over the step size of the integration path, which spans a much larger range than the local approximation considered here. Therefore, these are fundamentally different processes. Moreover, the infinitesimal nature of $r$ strictly confines the contribution of each dimension, ensuring a unique set of attributions for any given pair $\{x_1, x_2\}$. Consequently, one can leverage these pairs and the derived ground truth to rigorously assess whether different attribution methods produce valid attributions. This process requires only the storage of the corresponding samples $x_1$ and $x_2$, along with the attribution results constructed across different dimensions. The evaluation can then be performed by measuring the discrepancy between the attribution result generated by the algorithm on $x_1$ and the established ground truth.

## 4 CASE STUDY

### 4.1 MODEL AND BASELINES

Although our theoretical analysis is rigorous, infinitesimal concepts cannot be directly applied in practice due to finite numerical precision. It is important to emphasize that the correctness of our conclusions is guaranteed by theory, not by empirical observation. The experiments presented here should therefore be understood as a case study designed to illustrate how the theory manifests under

finite computational precision, rather than as a proof of the theory itself. This distinction also highlights that while the theoretical results are exact, practical implementations inevitably approximate them within machine precision.

To further investigate this, we provide an empirical example that evaluates the effect of numerical precision. In this experiment, we use the ResNet50 (He et al., 2016) model as the backbone for classification. We select nine interpretability methods, including Integrated Gradients (Sundararajan et al., 2017) (IG), Fast-Integrated Gradients (Hesse et al., 2021) (Fast-IG), Guided Integrated Gradients (Kapishnikov et al., 2021) (Guided-IG), Boundary-based Integrated Gradients (Wang et al., 2021) (BIG), SmoothGrad (Smilkov et al., 2017), Saliency Map (Simonyan et al., 2013) (SM), More Faithful and Accelerated Boundary-based Attribution (Zhu et al., 2024c) (MFABA), AttEXplore (Zhu et al., 2024b), and Iterative Search Attribution (Zhu et al., 2024a) (ISA). All experiments are conducted on an NVIDIA L40S GPU with Python 3.12.2. To ensure fairness, all experiments adopt an identical preprocessing pipeline, including clamping input values to $[0, 1]$ and applying ImageNet mean–std normalization before feeding into model. Thus, every attribution method operates under the same input conditions, enabling a fair comparison.

It is worth noting that only certain methods, such as Integrated Gradients and its variants (Fast-IG, Guided-IG, BIG), satisfy the attribution axioms defined in (Sundararajan et al., 2017). In contrast, a number of widely used approaches—including Grad-CAM (Selvaraju et al., 2017), Grad-CAM++ (Chattopadhay et al., 2018), DeepLIFT (Shrikumar et al., 2017), Layer-wise Relevance Propagation (LRP) (Bach et al., 2015), DeepSHAP/SHAP (Lundberg & Lee, 2017), and RISE (Petsiuk et al., 2018)—do not satisfy these axioms. Since our focus is on attribution methods that adhere to the axiomatic framework, we do not include these non-compliant methods in our experiments.

Nevertheless, even attribution methods that are axiomatically correct (e.g., IG) are not immune to risks: our theoretical framework demonstrates that trust-related vulnerabilities may persist regardless of whether the method satisfies attribution axioms. Hence, our experiments should be interpreted as a case study that verifies the compatibility of state-of-the-art attribution methods with our theoretical assumptions, while also revealing that similar risks emerge for other approaches. The phenomenon we identify is thus not confined to a specific method but reflects a broader issue that requires careful handling.

In this sense, our work calls for the interpretability community to pay closer attention to these risks and to develop practical solutions that explicitly address them, rather than assuming that the adoption of theoretically sound methods alone is sufficient.

We also employ the ImageNet dataset (Deng et al., 2009) in this paper. Following the setup in ISA (Zhu et al., 2024a), we randomly sample 1000 instances from ImageNet to evaluate the performance of various interpretability methods.

## 4.2 DISTANCE METRICS

To quantitatively analyze attribution differences, we employ three distance metrics. The Manhattan distance ($L_1$ norm) sums the absolute differences between corresponding elements, capturing the total variation in attribution maps. The Euclidean distance ($L_2$ norm) takes the square root of the sum of squared differences, reflecting the overall magnitude of deviation. Finally, the Cosine distance, defined as $1 - \cos(\theta)$ with $\cos(\theta)$ denoting the cosine similarity between two attribution vectors, measures directional differences in attributions rather than magnitude.

## 4.3 EXPERIMENT 1: CONSTRUCTING GROUND TRUTH

In this section, we refer to Eq. 3 to construct the ground truth, where $x_1$ and $x_2$ are iteratively updated using Eq. 1. Different learning rates $\eta$ are set, which are continuously reduced with the experiments. We first examine how decreasing the model's output magnitude (by scaling the input-output relationship) affects attribution accuracy. In this controlled setting, we treat attributions computed with an infinitesimal step as the ground truth and compare them against attributions obtained with finite step sizes $\eta$. We make infinitesimal approximations by continuously increasing the precision within the allowable precision range. At the same time, Table 1 also shows that the improvement in precision becomes increasingly smaller. Importantly, since all experiments are ultimately performed under finite machine precision, the conclusions drawn here remain valid and evident within such com-

Table 1: Attribution error vs. step size $\eta$ for a decreasing-output scenario (Exp. 1). Smaller $\eta$ yields lower distances, indicating more accurate (true) attributions.

| Metric | $\eta = 1/255$ | $\eta = 1/2550$ | $\eta = 1/25500$ |
|---|---|---|---|
| Manhattan Dist. | 7499.56 | 3020.18 | 976.19 |
| Euclidean Dist. | 42.89 | 19.68 | 7.18 |
| Cosine Dist. | 0.0489 | 0.0235 | 0.0068 |

Table 2: Distance between each method's attribution and the ground-truth attribution (Exp. 3). Lower values indicate higher fidelity to true attributions. Gradient-based methods (Fast/GIG, IG, SG, SM) have consistently low errors, whereas methods like BIG, MFABA, AttEXplore, ISA show large deviations.

| Metric | FIG | GIG | IG | SG | BIG | SM | MFABA | AttEXplore | ISA |
|---|---|---|---|---|---|---|---|---|---|
| Manhattan | 7638 | 7650 | 7653 | 7885 | 14692 | 7571 | 11223 | 11300 | 13385 |
| Euclidean | 42.09 | 42.98 | 42.57 | 43.98 | 74.75 | 42.49 | 59.98 | 58.77 | 73.88 |
| Cosine | 0.039 | 0.044 | 0.041 | 0.045 | 0.322 | 0.041 | 0.335 | 0.308 | 0.176 |

putational limits. In fact, when attribution differences shrink below machine precision, they also fall beneath human perceptual discrimination, making such numerical extremity the most effective practical evaluation approach.

When $\eta = 1/255$, it represents modifying a single pixel value in the image, whereas when $\eta = 1/2550$ or $\eta = 1/25500$, the modification to the image is even smaller than a single pixel value. Table 1 summarizes the attribution error (distance between true and estimated attributions) for three decreasing step sizes. As the output is scaled down, using a smaller $\eta$ leads to dramatically smaller attribution errors across all metrics.

As shown in Table 1, with $\eta = 1/255$, the Manhattan distance between the estimated and true attributions is approximately 7499. However, when the step size is reduced by a factor of ten to $\eta = 1/2550$, the Manhattan distance drops to around 3020, and with $\eta = 1/25500$, it further decreases to about 976. We observe a similar monotonic decrease in Euclidean distance (from 42.89 to 7.18) and Cosine distance (from 0.0489 to 0.0068) as $\eta$ shrinks.

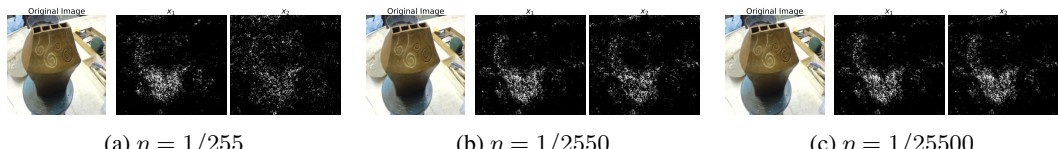

(a) $\eta = 1/255$  (b) $\eta = 1/2550$  (c) $\eta = 1/25500$

Figure 2: Attribution results for different $\eta$. As $\eta$ decreases, the attribution results become increasingly consistent (Attributions of $x_1$ and $x_2$) become same.

From Figure 2, we can see that when the initial step size $\eta$ is relatively large, the attribution results exhibit noticeable differences. However, as $\eta$ decreases, the attribution results become almost entirely consistent. From a results perspective, the phenomenon discussed in this work becomes clearly evident when $\eta = 1/25500$, suggesting that this level of precision is sufficient as a constraint in practice, particularly given the finite numerical resolution of computer implementations.

## 4.4 EXPERIMENT 2: ATTRIBUTION SIMILARITY ACROSS CASES

In this experiment, we apply nine different attribution algorithms to analyze two highly similar images, $x_1$ and $x_2$, which belong to different classes (For example, $x_1$ belongs to class $A$, $x_2$ belongs to class $B$). As shown in Figure 3, the attribution results for $x_1$ and $x_2$ are highly similar across all methods. This validates the concern we raised earlier: when attribution algorithms process two infinitesimally close samples that belong to different categories, they can produce nearly identical attribution results.

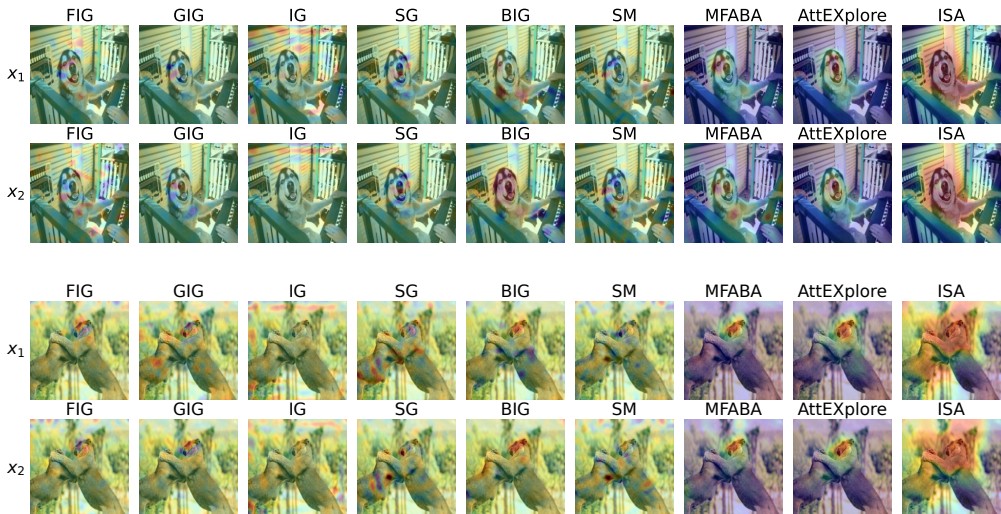

Figure 3: Attribution results for two infinitesimally close samples, $x_1$ and $x_2$, across different attribution methods for different sample indices.

### 4.5 EXPERIMENT 3: TRUSTWORTHINESS OF ATTRIBUTION METHODS

In the final experiment, we directly evaluate each method's trustworthiness by comparing its attributions to the ground-truth attribution. Here, the ground truth is defined as the attribution obtained under the most stringent setting (the lowest $\eta$ or essentially an oracle with minimal approximation error). We compute the distance between each method's attribution and this true attribution (for the same input and output). Table 2 reports these distances, and Figure 3 visualizes them. The results reveal clear discrepancies in how faithfully different methods recover the true attribution. Standard gradient-based methods – FIG, GIG, vanilla IG, SG, and the method SM – all stay very close to ground truth (Cosine distances $\approx 0.04$, Euclidean $\approx 42$, Manhattan around $7.6 \times 10^3$). In fact, their cosine similarity to the true attribution exceeds 0.96 (since Cosine distance $\approx 0.04$), indicating that these methods reliably identify the correct features and their importance.

In contrast, several methods deviate markedly from the true attributions. BIG, MFABA, and AttEXplore show much larger errors. BIG is furthest from the truth, with a Cosine distance of 0.322 (cosine similarity $\sim 0.68$) and a Manhattan error of 14,692—about twice that of IG (7653). MFABA and AttEXplore also perform poorly, with Cosine distances of 0.31–0.34 (an order of magnitude worse than IG/SG) and Euclidean distances ($\approx 59$–75), about 1.5–1.8× higher than reliable methods. ISA underperforms IG as well, with Cosine 0.176 (4× worse than IG) and Manhattan 13,385. These results highlight that some attribution techniques substantially diverge from the truth. Figure 4 in **Appendix A** illustrates this: methods like FIG, GIG, IG, SG, and SM have near-zero error bars, while BIG, MFABA, and AttEXplore reach high values. Practically, relying on BIG-like methods risks misleading users about key features. Experiment 2 thus confirms that not all attribution methods are trustworthy—IG and SG are more reliable, whereas others require caution or improvement to align with ground truth.

## 5 CONCLUSION

In this work, we highlight a trust issue in attribution tasks, where even perfect algorithms can fail to distinguish key decision boundaries, causing ambiguity in feature importance. Infinitesimally close samples with different classifications can share the same attribution region, undermining reliability. This issue points to a core limitation of the attribution framework, suggesting future research must address inherent uncertainties in model explanations. While our method provides valuable insights, its applicability in non-classification tasks remains limited, and alternative methods should be explored to better handle these uncertainties.

## ETHICS STATEMENT

We have read and will adhere to the ICLR Code of Ethics. This work uses only public data, involves no human subjects or personally identifiable information, and therefore does not require IRB review. Results are reported for research purposes only; we release anonymized code/configurations to support verification, and will disclose any funding sources and potential conflicts of interest upon acceptance.

## REPRODUCIBILITY STATEMENT

To support reproducibility, we release an anonymized repository with all experiment details including training/evaluation scripts, default hyperparameters, configuration files, and software/hardware environment.

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

## LLM USAGE DISCLOSURE

We used large language models (OpenAI GPT-4o and GTP-5) as auxiliary tools for grammar checking and language polishing of the manuscript. These models were not involved in research ideation, experimental design, implementation, or analysis. The authors take full responsibility for all content.

## A  ATTRIBUTION EVALUATION RESULTS FOR EXPERIMENT 3

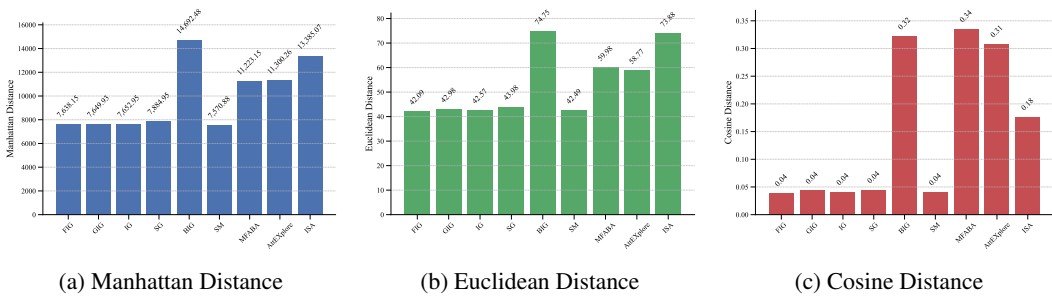

(a) Manhattan Distance                (b) Euclidean Distance                (c) Cosine Distance

Figure 4: Deviation from Ground-Truth Attributions using different distance metrics. Lower values indicate higher attribution faithfulness. FIG, GIG, IG, SG, and SM exhibit minimal deviations, while BIG, MFABA, and AttEXplore have significantly higher errors, with ISA showing moderate deviation.

## B  ADDITIONAL VISUALIZATION OF THE ATTRIBUTION DISTRUST ISSUE

In this appendix, we additionally show attribution visualizations for different pairs of input samples $x_1$ and $x_2$. From additional visualization results, we can draw the same conclusion as presented in Fig. 3 of the main text: for two infinitesimally close samples belonging to different classes, attribution methods tend to assign importance to the same feature attention regions.

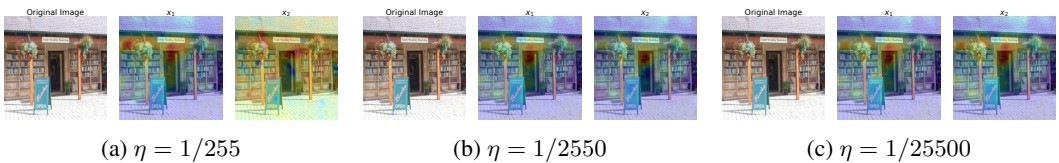

(a) $\eta = 1/255$                (b) $\eta = 1/2550$                (c) $\eta = 1/25500$

Figure 5: Additional attribution results for different $\eta$. As $\eta$ decreases, the attribution results become increasingly consistent (Attributions of $x_1$ and $x_2$ become the same).

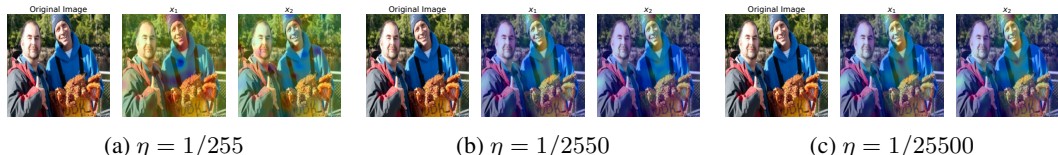

(a) $\eta = 1/255$                (b) $\eta = 1/2550$                (c) $\eta = 1/25500$

Figure 6: Additional attribution results for different $\eta$. As $\eta$ decreases, the attribution results become increasingly consistent (Attributions of $x_1$ and $x_2$ become the same).

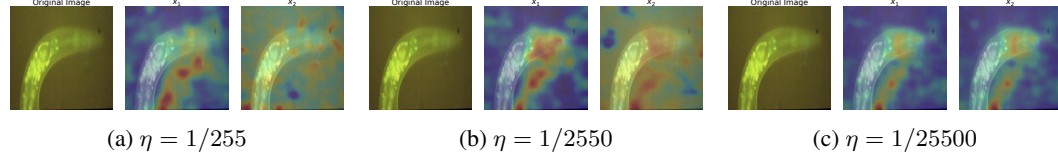

(a) $\eta = 1/255$    (b) $\eta = 1/2550$    (c) $\eta = 1/25500$

Figure 7: Additional attribution results for different $\eta$. As $\eta$ decreases, the attribution results become increasingly consistent (Attributions of $x_1$ and $x_2$ become the same).

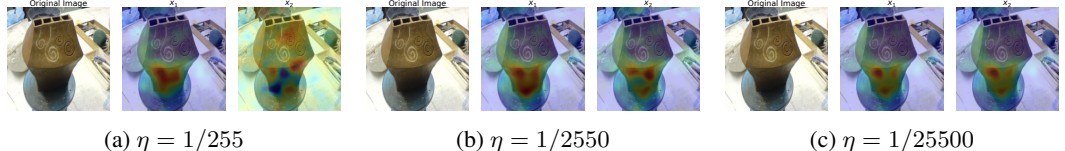

(a) $\eta = 1/255$    (b) $\eta = 1/2550$    (c) $\eta = 1/25500$

Figure 8: Additional attribution results for different $\eta$. As $\eta$ decreases, the attribution results become increasingly consistent (Attributions of $x_1$ and $x_2$ become the same).

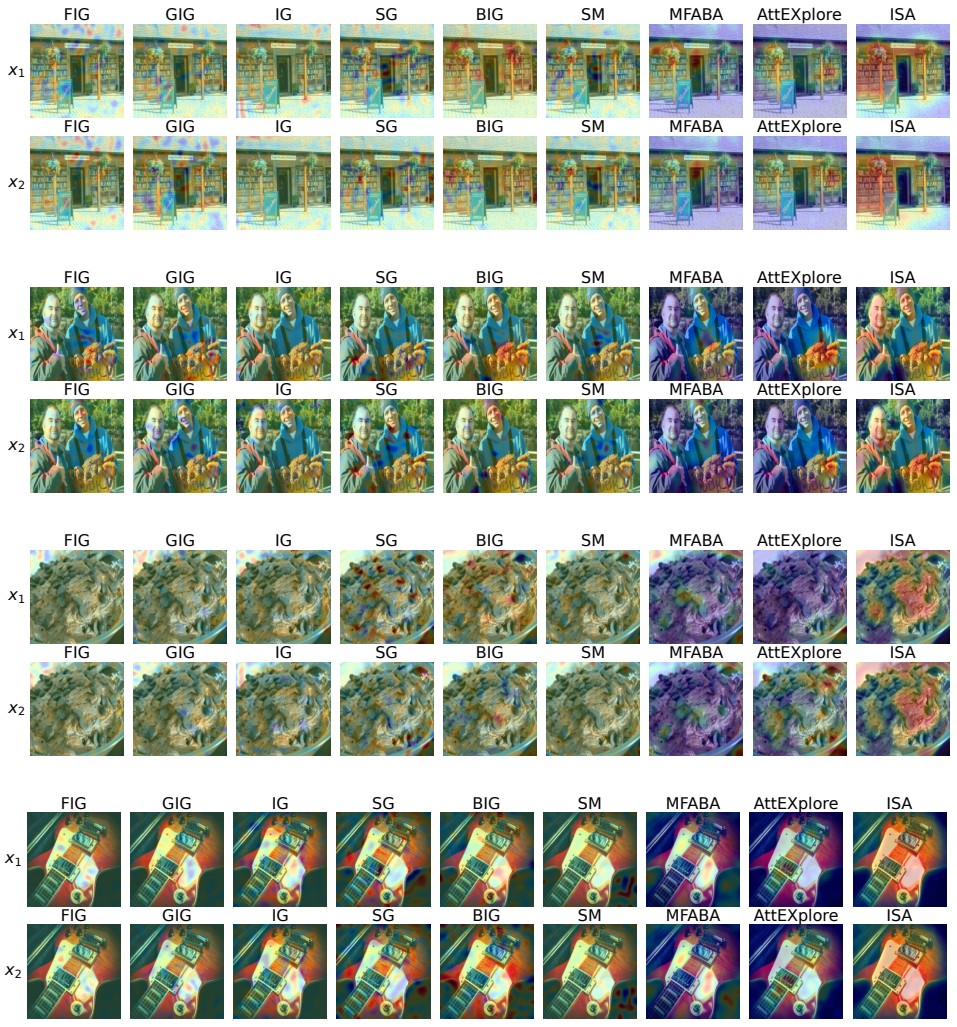

Figure 9: Additional attribution results for two infinitesimally close samples, $x_1$ and $x_2$, across different attribution methods for different sample indices.

