# OpenReview forum: "Can we trust the attribution method?"
_ICLR.cc/2026/Conference — ICLR 2026 Conference Withdrawn Submission_

### Official Review · Reviewer_HM8k · 2025-10-29

**Soundness:** 1
**Presentation:** 2
**Contribution:** 1
**Rating:** 2
**Confidence:** 4

**Summary:**

The paper studies whether feature attributions are trustworthy. It highlights a near-boundary phenomenon that two almost identical inputs with opposite predictions can yield very similar saliency, and proposes a “ground-truth attribution” defined by decomposing the output change between the two inputs along the input-space path. Using this construction, it compares multiple attribution methods and argues that satisfying axioms does not guarantee reliability.

**Strengths:**

1. The trustworthiness of attribution methods is well-motivated and directly relevant to ongoing debates in explainability.

2. The problem setup and the central question are clearly articulated.

**Weaknesses:**

1. This work has a faulty “ground truth” premise. The paper assumes the output change should be fully or primarily attributed to the pixels that differ between the two images. In nonlinear networks, small pixel perturbations can redistribute importance globally. Therefore, input difference is not a faithful surrogate for measuring the reliability of attributions. The completeness axiom constrains attribution to sum to output change, not to mirror input interpolation. Hence, the proposed “ground truth” cannot validly measure attribution fidelity, making the subsequent experiments and conclusions problematic.

2. The claim that “satisfying axioms does not ensure reliability” adds limited novelty. Attribution axioms are widely regarded as necessary but not sufficient.

3. Robustness and reliability of explanations have been examined in many existing works[1,2,3]. In contrast, this work brings few new conclusions.

[1] Rudin, Cynthia. "Stop explaining black box machine learning models for high stakes decisions and use interpretable models instead." Nature machine intelligence 2019.

[2] Zhou, Yilun, et al. "Do feature attribution methods correctly attribute features?." AAAI 2022.

[3] Dombrowski, Ann-Kathrin, et al. "Explanations can be manipulated and geometry is to blame." Advances in neural information processing systems 2019.

**Questions:**

1. How do you justify that the output change should be fully (or primarily) attributed to the perturbed pixels, despite nonlinear cross-feature interactions?
2. Why must two near-boundary images necessarily have different attributions? If you cannot establish this necessity, on what basis do you reject the reliability of existing methods rather than concluding that visual similarity alone is inconclusive?
3. What new conclusions or insights does your evaluation add beyond existing studies?

---

### Official Review · Reviewer_oFag · 2025-10-31

**Soundness:** 1
**Presentation:** 3
**Contribution:** 2
**Rating:** 2
**Confidence:** 4

**Summary:**

This paper investigates an intrinsic trust limitation of feature attribution methods, arguing that even a theoretically perfect algorithm that yields the “ground-truth attribution” can still be untrustworthy: for two samples that are infinitesimally close yet classified into different labels, their attributions may highlight the same “important” region, thus failing to capture what truly distinguishes the decisions.

However, the paper suffers from a fundamental conceptual flaw. The authors conflate classification explanation with contrastive attribution. By construction, the attributions of two nearby samples near the decision boundary are expected to be similar under the local smoothness assumption. In such cases, meaningful analysis should focus on the differential change ($\Delta$ attribution) that explains the class transition, rather than interpreting attribution similarity as a failure of trustworthiness. Consequently, the core argument misinterprets the nature of attribution and the semantics of decision explanation.

My suggestion is to reframe the comparison to $\Delta$ attribution and see if the trustworthiness are violated in that setup.

**Strengths:**

1. The paper raises an important and underexplored question: whether attribution methods can ever be fully trustworthy even when theoretically correct.
2. The authors provide a mathematically grounded setup for analyzing infinitesimally close sample pairs across decision boundaries.

**Weaknesses:**

1. The proposed “ground-truth attribution” derived from the residual–gradient coupling fundamentally attributes class transition rather than class membership. In other words, it measures why the prediction changes from class A to class B, not why the model predicts class A. This conflates attribution with adversarial sensitivity, departing from the standard semantics of attribution methods such as IG or SHAP, which are designed to decompose a single class score into feature contributions.

2. Because the ground truth is inherently contrastive (A-->B), comparing it against non-contrastive attributions (e.g., IG(x)) creates an unfair benchmark. To evaluate the behavior of traditional attribution methods under this setup, one should instead compare their attribution differentials, such as $\Delta IG = IG(x) − IG(x′)$, rather than the absolute maps themselves. Otherwise, the observed “similarity” between two attributions merely reflects local smoothness rather than a genuine failure of trustworthiness.

reference: Pan, Deng, Xin Li, and Dongxiao Zhu. "Explaining deep neural network models with adversarial gradient integration." Thirtieth International Joint Conference on Artificial Intelligence (IJCAI). 2021.

**Questions:**

Your “ground-truth attribution” is defined through two similar samples belonging to different classes. Does this formulation aim to explain why the model predicts class A or why the prediction changes from A to B? Please clarify whether you regard this as a classification or contrastive explanation, and how it aligns with standard attribution semantics (e.g., IG, SHAP).

---

### Official Review · Reviewer_wofT · 2025-11-01

**Soundness:** 2
**Presentation:** 3
**Contribution:** 2
**Rating:** 4
**Confidence:** 4

**Summary:**

This paper investigates a fundamental limitation in attribution. The authors demonstrate those method gives misleading trust of attribution through scenarios where two nearly identical inputs yield different classifications but share the same attribution. This shows that attribution methods may fail to distinguish decisive features in classification shifts. The authors offer theoretical foundation for this phenomenon and propose a benchmark for evaluating attribution reliability. They caution that attribution results should not be blindly trusted, even when the method is technically correct.

**Strengths:**

- The paper is well-written and easy to follow.
- The paper points out an interesting question regarding the trust of attribution, and provides rigorous analysis to the question.

**Weaknesses:**

1. Can we indeed construct such $x_1$ and $x_2$? According to my understanding, the existence of $x_1$ and $x_2$ is based on the fact that they lies closely to the decision boundary, such that $\forall\varphi>0$, $\exists x_{1}, x_{2}\in[\gamma(t)\mid t \in (0, 1)]$, satisfying $C(x_1) \neq C(x_2)$ and $\|x_1 - x_2\| < \varphi$. However, is there any guarantee that those samples are semantically meaningful images?
2. $\alpha(t)$ is not defined. Actually, what is the purpose of $\alpha(t)$.
3. I have concerns that if it is intrinsically supposed to be the phenomenon that nearly identical inputs could have same attributions? For example, if we consider two images with all black but one pixel different (e.g., one red and one blue). The classifier classifies them differently (red and blue). And the attribution should be able to identify that pixel and the attribution region should be the same.

**Questions:**

See weaknesses.

---

### Official Review · Reviewer_BhXD · 2025-11-03

**Soundness:** 1
**Presentation:** 1
**Contribution:** 2
**Rating:** 2
**Confidence:** 4

**Summary:**

This paper raises a trust issue in feature attribution--two samples with infinitesimal differences but different predicted labels can have the same attribution map. A theoretical result is developed to demonstrate this issue. Empirical results on image classification with ResNet show that this theoretical issue can occur in practice. Overall, this paper challenges the assumption that axiom-justified attribution methods can always be trusted.

**Strengths:**

- This paper proposes an original issue related to trustworthiness in feature attribution. Previous work such as [R1] and [R2] typically shows the issue that similar samples can have different explanations, while this work shows that similar samples with different predicted labels can have the same explanation.

References

[R1] Dombrowski et al. (2019) - Explanations can be manipulated and geometry is to blame

[R2] Slack et al. (2021) - Feature Attributions and Counterfactual Explanations Can Be Manipulated

**Weaknesses:**

- In the paper, there are two ways of evaluating the trustworthiness of an attribution method. (1) A trustworthy attribution method should capture the ground truth about the explained model's decision-making process. This is the evaluation criterion in Section 4.5. (2) A trustworthy attribution method should have different attribution maps for two similar samples with different predicted labels. This is the evaluation criterion introduced by the paper. In the setting considered in this paper, (1) and (2) contradict each other. It then becomes unclear why (2) is more relevant to user trust than (1). Can't we argue that an attribution method that satisfies (1) but fails (2) is trustworthy because the method reveals counterintuitive aspects about a model's decision-making process, highlighting that the model itself or the samples are not trustworthy?

- The proof for Theorem 1 seems incomplete. Specifically, the property $\lVert x_1 - x_2 \rVert \le \phi$ is not (directly) proven.

- In line 231, it is stated that "as $\epsilon \rightarrow 0, \lVert x_1 - x_2 \rVert$ approaches 0." This does not hold directly from the continuity of a neural network (as stated in line 226), where the implication is in the opposite direction. I think the authors can remove this statement as currently it is not required to prove line 232.

- In lines 269-270, it is argued that $r = x_2 - x_1$ transitions the model's output from class A to class B, and that is the basis for treating Equation 3 as the ground truth attribution. However, since $x_2$ is not unique (as mentioned in lines 234-244), it's possible to have another adversarial sample $x_3$ such that $r' = x_3 - x_1$ is small and transitions the model's output from class A to class B. Since $r' \neq r$, applying Equation 3 gives us two ground truths. It then becomes unclear what the notion of ground truth is.

**Questions:**

- Based on the exposition in the paper, an attribution method that captures the ground truth is not trustworthy because it produces identical attribution maps for similar samples that have different predicted labels. Can't we argue that the attribution method itself is trustworthy because it reveals counterintuitive aspects about the explained model's decision-making process, highlighting that the model itself or the samples are not trustworthy.

- It seems like the scenario where two similar samples have different predicted labels arise only from adversarial samples. In this case, can't we argue an attribution method that fails the trustworthiness criterion proposed in lines 60-61 is actually trustworthy, because the method reveals counterintuitive aspects about the adversarial samples?

- Related to the second point, is it possible to construct samples $x_1$ and $x_2$ without adversarial attack? In other words, can $x_1$ and $x_2$ be sampled from $P(x)$ directly? What assumptions about $P(x)$ might be necessary, and are they reasonable?

- Currently, the last property $\lVert x_1 - x_2 \rVert \le \phi$ in Theorem 1 is not (directly) proven. Can you provide the completed proof?

- Since there can be multiple adversarial samples that flip the predicted label in the same direction, are there multiple ground truth attribution maps?

- Section 4.4 only shows qualitative visual examples. Can you provide quantitative comparisons between different methods? Is there a tradeoff between the trustworthiness criterion in Section 4.4 vs. the faithfulness criterion in Section 4.5?

---

### Note · Authors · 2025-11-14

I have read and agree with the venue's withdrawal policy on behalf of myself and my co-authors.